# Key Roles of Brown, Subcutaneous, and Visceral Adipose Tissues in Obesity and Insulin Resistance

**DOI:** 10.3390/cimb47050343

**Published:** 2025-05-09

**Authors:** Maria-Zinaida Dobre, Bogdana Virgolici, Olivia Timnea

**Affiliations:** 1Department of Biochemistry, Faculty of Medicine, Carol Davila University of Medicine and Pharmacy, 050474 Bucharest, Romania; maria.dobre@umfcd.ro; 2Department of physiology, Faculty of Sports and Kinetotherapy, Romano-Americana University, 012101 Bucharest, Romania; oliviatimnea@yahoo.com

**Keywords:** adipose tissue, insulin resistance, obesity, visceral fat, subcutaneous fat, brown adipose tissue, inflammation, mitochondria, adipokines, metabolic flexibility, adipose progenitor cells

## Abstract

Adipose tissue is a dynamic and heterogeneous organ with distinct depots that play divergent roles in metabolic regulation. This review highlights the functional differences between brown, subcutaneous, and visceral adipose tissue, and their contributions to obesity-related insulin resistance. We explore how chronic low-grade inflammation, mitochondrial dysfunction, and fibrosis evolve within specific fat depots and how these changes disrupt systemic energy homeostasis. Visceral white adipose tissue (vWAT) emerges as a critical site of inflammation and metabolic inflexibility, while subcutaneous white adipose tissue (sWAT) may retain protective features in early obesity. The endocrine roles of adipokines and batokines are also discussed, emphasizing depot-specific signaling and systemic effects. Furthermore, we examine emerging therapeutic strategies aimed at modulating immune responses, enhancing mitochondrial function, and reprogramming adipose progenitor cells (APCs) to restore healthy tissue remodeling. A deeper understanding of adipose-depot-specific biology and progenitor cell dynamics offers promising avenues for personalized interventions in metabolic diseases.

## 1. Introduction

### 1.1. Obesity and Insulin Resistance as a Disease

Obesity and insulin resistance represent critical global health challenges, fueling the rising incidence of metabolic disorders, such as type 2 diabetes, cardiovascular disease, and metabolic-dysfunction-associated fatty liver disease (MAFLD). According to the World Health Organization (WHO), over 2.5 billion adults were overweight in 2022, including more than 890 million living with obesity. Alarmingly, pediatric obesity is also escalating, with an estimated 37 million children under 5 and over 390 million aged 5–19 classified as overweight [1]. These statistics underline the growing burden of obesity across all age groups and socioeconomic backgrounds.

### 1.2. Key Contributing Factors to Obesity and Insulin Resistance

The surge in obesity prevalence is driven by a complex interplay of genetic predisposition, sedentary behavior, excessive caloric intake, and environmental influences [2]. Insulin resistance, defined as a diminished biological response of key target tissues, such as liver, muscle, and adipose tissue to normal or elevated circulating insulin levels, plays a central role in the development of obesity-related metabolic diseases [3]. This dysfunction impairs glucose uptake and utilization, promotes hepatic glucose production, and contributes to systemic hyperglycemia. While the association between obesity and insulin resistance is well-established, the specific contributions of different adipose tissue depots to metabolic dysfunction remain incompletely understood. Importantly, recent research has revealed that adipose tissue acts not merely as an energy reservoir but as an active endocrine organ with depot-specific metabolic, inflammatory, and cellular characteristics, including mitochondrial dysfunction and the aberrant behavior of adipose progenitor cells (APCs).

### 1.3. Outline of the Review Themes

Among the various adipose depots, visceral white adipose tissue (vWAT) has been implicated as a major driver of systemic insulin resistance through its heightened lipolytic activity, pro-inflammatory cytokine secretion, and immune cell infiltration [4]. In contrast, subcutaneous white adipose tissue (sWAT) has traditionally been considered metabolically neutral or protective, though recent evidence highlights the detrimental role of sWAT fibrosis in insulin resistance, particularly when expandability is limited [5]. Brown adipose tissue (BAT) and beige adipose tissue (BeAT) have gained attention for their thermogenic and insulin-sensitizing properties; however, the effectiveness of BAT activation in severe insulin resistance remains debated [6], and the functional stability of BeAT under obesogenic conditions is still uncertain [7]. Additionally, the plasticity of beige adipocytes and the role of APCs in depot-specific remodeling and fibrosis represent critical areas of investigation [8].

This review aims to provide a nuanced understanding of adipose tissue heterogeneity by exploring how distinct depots—sWAT, vWAT, BeAT, and BAT—contribute differently to metabolic outcomes. We integrate molecular and clinical perspectives, focusing particularly on the mitochondrial function, inflammatory signaling, and the therapeutic potential of targeting APCs to mitigate obesity-induced metabolic dysfunction.

## 2. Overview of BAT, BeAT, sWAT, and vWAT

### 2.1. Brown Adipose Tissue (BAT)

Brown adipose tissue (BAT) is a highly specialized form of adipose tissue characterized by its thermogenic capacity, which is primarily attributed to its high mitochondrial density and the expression of uncoupling protein 1 (UCP1) localized in the inner mitochondrial membrane [9,10]. Upon cold exposure or β3-adrenergic stimulation, sympathetic neurons release norepinephrine, which activates β3-adrenergic receptors on brown adipocytes. This triggers a signaling cascade involving cyclic AMP (cAMP), protein kinase A (PKA), and the transcriptional coactivator PGC-1α, leading to mitochondrial biogenesis, UCP1 upregulation, and a subsequent proton leak across the mitochondrial membrane, dissipating energy as heat instead of producing ATP [10,11,12].

The discovery of metabolically active BAT in adult humans, confirmed by PET-CT imaging studies in 2009, marked a paradigm shift in our understanding of thermogenesis and energy regulation [11,13,14]. Beyond thermogenesis, BAT contributes to systemic metabolic health by improving glucose metabolism, increasing energy expenditure, and reducing fat accumulation [15,16]. Recent findings [17] further associated the presence of BAT with improved cardiometabolic profiles in humans, suggesting a protective role against obesity-related complications.

Given its significant impact on the energy balance and metabolism, BAT has emerged as an attractive therapeutic target for combating obesity and insulin resistance. However, BAT’s function and prevalence vary considerably between individuals, influenced by factors such as age, sex, and metabolic status [6,13]. In obese individuals, BAT activation may not confer the same metabolic benefits observed in lean individuals. Moreover, under certain conditions, like chronic nutrient overload or hypoxia, BAT may recruit pro-inflammatory immune cells, contributing to low-grade inflammation and potentially limiting its therapeutic applications [6,18].

Additionally, while the role of adipose progenitor cells (APCs) in BAT development and plasticity remains poorly understood, emerging evidence suggests that their interplay with BAT’s inflammatory response could critically influence brown fat functionality [18,19]. Thus, a deeper understanding of BAT biology, including its immunometabolic interactions, is essential to fully exploit its therapeutic potential.

### 2.2. Beige Adipose Tissue (BeAT)

Beige adipose tissue (BeAT), also known as inducible brown fat, represents a metabolically flexible form of adipose tissue that can emerge within sWAT in response to cold exposure, exercise, or pharmacological stimulation [20]. BeAT shares thermogenic properties with BAT but differs in its cellular origins and regulatory mechanisms. Unlike BAT, which is developmentally distinct, BeAT forms through the the transdifferentiation of white adipocytes into thermogenically active beige cells. While BeAT has been associated with improved metabolic health, its ability to sustain thermogenesis in an obesogenic environment remains uncertain [21]. Additionally, recent findings suggest that beige adipocytes may exhibit pro-inflammatory potential under metabolic stress, similar to BAT, warranting further investigation into their long-term effects on systemic metabolism [22].

### 2.3. Subcutaneous White Adipose Tissue (sWAT)

Subcutaneous white adipose tissue (sWAT) is traditionally regarded as a metabolically protective fat depot. However, its function becomes compromised under metabolic stress, undergoing fibrotic remodeling and immune cell infiltration. In sWAT, adipose progenitor cells (APCs) play a critical role in maintaining tissue plasticity by enabling healthy adipogenesis. Recent advances using single-cell and single-nucleus RNA sequencing have uncovered depot-specific immune and stromal cell subpopulations that influence sWAT homeostasis. In particular, some studies [23,24] have mapped the heterogeneity of APCs and immune cells in sWAT under lean and obese conditions. These findings are supported by Bäckdahl et al. (2021) [25], who highlighted the impact of the extracellular matrix on progenitor fate, and by Gupta et al. (2022) [26], who emphasized the advantages of single-nucleus sequencing in studying mature adipocytes. Importantly, confusion between “sWAT” (subcutaneous white adipose tissue) and “SWAT cells” (a structural Wnt-regulated progenitor population identified in bone marrow) should be avoided, as the two are unrelated in function and origin.

### 2.4. Visceral White Adipose Tissue (vWAT)

Visceral white adipose tissue (vWAT) is the most metabolically active and pro-inflammatory adipose depot that is strongly linked to insulin resistance, metabolic syndrome, and cardiovascular disease [12]. Unlike sWAT, vWAT is highly infiltrated by pro-inflammatory M1 macrophages, which sustain chronic low-grade inflammation. This inflammatory environment, exacerbated by adipocyte hypertrophy and hypoxia, perpetuates insulin resistance and metabolic dysfunction [27].

Beyond inflammation, recent research highlights the role of adipose progenitor cells (APCs) in vWAT pathology. While vWAT expansion is typically associated with hypertrophy and fibrosis, certain APC subpopulations retain regenerative potential and contribute to tissue remodeling [4]. These findings highlight the complex interplay between immune cells and APCs in vWAT, shifting the focus beyond inflammation alone and toward a broader understanding of adipose tissue remodeling.

Together, these insights emphasize the need for a paradigm shift in adipose tissue research. Rather than categorizing depots as simply harmful or protective, their roles should be viewed as context-dependent. Understanding the interactions between immune cells and APCs in both sWAT and vWAT is crucial for developing targeted therapies to combat obesity-related metabolic diseases. An overview of the key features of these adipose tissues is presented in Table 1.

## 3. Adipose Progenitor Cells and Fibrosis in Obesity

Adipose progenitor cells (APCs), also known as adipose-derived stem cells (ADSCs), are pivotal in the formation, expansion, and functional adaptation of adipose tissue. These multipotent progenitors orchestrate adipose tissue remodeling in response to metabolic stressors, such as obesity [28]. In obesity, APC differentiation is profoundly altered, with depot-specific behaviors contributing to the adipose tissue heterogeneity and metabolic dysfunction [29]. In sWAT, APCs primarily differentiate into adipocytes, supporting healthy tissue expansion and lipid storage. However, in obesity, chronic inflammation and mechanical stress impair the APC function, prompting their differentiation into fibroblast-like cells and fostering excessive extracellular matrix (ECM) deposition, leading to fibrosis [30].

Fibrosis, a maladaptive response to excessive adipose tissue expansion, reduces tissue plasticity and impairs metabolic function. In lean individuals, APCs in sWAT undergo hyperplasia, generating new adipocytes to accommodate lipid surplus and prevent ectopic fat deposition. In contrast, obesity-induced chronic inflammation and mechanical stress shift APC differentiation toward fibrosis, which impairs tissue expandability and induces hypertrophic growth [31]. Hypertrophic adipocytes are more susceptible to hypoxia, oxidative stress, and inflammatory cytokine production, all of which exacerbate metabolic dysfunction and worsen systemic metabolic health [32].

The functional state of APCs differs significantly between lean and obese individuals, as presented in Figure 1. In lean states, APCs maintain a balanced differentiation potential, generating adipocytes in response to caloric excess while remaining responsive to anti-inflammatory and insulin-sensitizing signals [33]. In obesity, however, APC differentiation is dysregulated by increased exposure to pro-inflammatory cytokines (e.g., TNF-α, IL-6) and fibrotic signals, such as transforming growth factor-beta (TGF-β). These factors suppress adipogenesis and promote myofibroblast-like differentiation, resulting in ECM rigidity and fibrosis [34]. Previous studies have highlighted that transforming growth factor-beta (TGF-β) plays a central role in promoting adipose tissue fibrosis by directing adipose progenitor cells (APCs) toward a myofibroblastic phenotype. Upon chronic exposure to inflammatory cues and mechanical stress, TGF-β signaling upregulates profibrotic genes, such as collagen type I (COL1A1) and alpha-smooth muscle actin (α-SMA), leading to excessive extracellular matrix (ECM) deposition and a loss of adipose tissue flexibility [35]. In addition to altered differentiation, autophagy and mitophagy dysfunctions in APCs significantly contribute to fibrogenesis. Impaired autophagic flux leads to the accumulation of dysfunctional mitochondria, increased oxidative stress, and the activation of senescence-associated secretory phenotype (SASP), which further propagates local inflammation and fibrotic remodeling [36]. Senescent APCs secrete pro-inflammatory cytokines, matrix-remodeling enzymes, and growth factors that stiffen the ECM and impair tissue expandability. This pathological stiffening feeds back to further alter the APC fate and function, locking adipose tissue in a maladaptive fibrotic state that diminishes its lipid-buffering capacity and exacerbates systemic insulin resistance [37]. Thus, the interplay between TGF-β signaling, defective autophagy, and cellular senescence in APCs represents a critical axis driving adipose tissue fibrosis and metabolic dysfunction in obesity. Obesity-associated APCs also exhibit impaired cellular functions, including reduced proliferation and differentiation capacity, disrupted mitochondrial morphology, decreased mitochondrial membrane potential, and diminished antioxidative and metabolic activities [38]. Moreover, they enhance the M1-type macrophage polarization while inhibiting M2-type polarization, further exacerbating local inflammation. Lysosomal dysfunction, characterized by accumulated undegraded materials, increased lysosomal permeability, and defective autophagy, contributes to the senescent phenotype of these cells. Transcriptomic analysis further highlights the upregulation of pathways linked to senescence, inflammation, and cancer, while those related to stemness, differentiation, and metabolic adaptation are downregulated [39]. This disruption not only impacts the adipogenic capacity but may also contribute to the proangiogenic features of obesity-derived APCs, as seen in their increased IL-6 expression and activation of Notch signaling pathways, further linking chronic inflammation to adipose tissue dysfunction [40]. Additionally, obesity-related hypoxia and mechanical stress further impair the APC function, reinforcing maladaptive tissue remodeling and contributing to the development of metabolic dysfunction [41].

### 3.1. Therapeutic Potential of Targeting APCs to Modulate Adipose Tissue Function

Adipose progenitor cells (APCs) represent an emerging therapeutic target for addressing obesity and related metabolic disorders given their critical role in adipose tissue homeostasis, differentiation, and fibrotic remodeling. Several strategies have been proposed to modulate APC function, aiming to restore healthy adipose tissue expansion, reduce fibrosis, and improve insulin sensitivity. These approaches focus on either enhancing the adipogenic differentiation or reducing the fibrotic and inflammatory microenvironment that hinders adipocyte formation [42].

### 3.2. Modulating ECM Remodeling to Reduce Fibrosis

Excessive extracellular matrix (ECM) deposition is a hallmark of adipose tissue fibrosis, which impairs tissue expandability and contributes to insulin resistance [43]. Targeting ECM remodeling presents a promising strategy to modulate APC function. One potential approach is the inhibition of transforming growth factor-beta (TGF-β) signaling, a key driver of fibrotic responses in adipose tissue. TGF-β promotes the differentiation of APCs into myofibroblast-like cells, which produce excessive ECM components, leading to fibrosis. Preclinical studies have shown that TGF-β inhibitors can reduce ECM deposition and improve adipogenesis in both sWAT and vWAT. These findings suggest that targeting TGF-β signaling could help restore healthy adipose tissue function and improve metabolic outcomes [44].

Matrix metalloproteinases (MMPs), which degrade ECM components, represent another therapeutic target. By promoting ECM turnover, MMPs could facilitate APC differentiation and prevent excessive fibrosis. Recent studies explored the use of MMP activators or small molecules to enhance ECM remodeling, demonstrating improvements in adipose tissue expandability and insulin sensitivity in animal models of obesity [45].

### 3.3. Targeting Inflammatory Pathways to Promote Healthy APC Function

Chronic inflammation plays a central role in obesity-related metabolic dysfunction by inhibiting adipogenic differentiation and promoting fibrosis. APCs are highly responsive to inflammatory signals, which can shift their fate from adipogenesis to fibrogenesis. Targeting inflammation through immune modulation represents a promising therapeutic strategy. One approach involves the use of anti-inflammatory agents, such as interleukin-10 (IL-10) or IL-4 mimetics, which have been shown to promote a shift toward an anti-inflammatory microenvironment in adipose tissue [46]. These agents can enhance APC differentiation into adipocytes by modulating macrophage polarization and reducing the production of pro-inflammatory cytokines [47]. For example, IL-10 has been shown to reduce the TNF-α levels in adipose tissue, thereby restoring insulin sensitivity and promoting healthy adipose tissue expansion. Another strategy involves the selective modulation of immune cell populations. For instance, the polarization of M1 macrophages, which are pro-inflammatory, to an M2 anti-inflammatory phenotype has been proposed as a means of improving APC function and adipose tissue homeostasis. Drugs or small molecules that favor the expansion of Tregs (regulatory T cells) in adipose tissue could also help restore the anti-inflammatory environment necessary for proper adipogenesis [48].

### 3.4. Pharmacological Activation of Adipogenesis

Adipogenesis plays a fundamental role in preserving metabolic health by enabling adipose tissue to safely store excess lipids, thereby preventing ectopic fat accumulation in organs such as the liver and muscle. In physiological conditions, healthy adipogenesis through progenitor cell differentiation leads to hyperplasia rather than hypertrophy, allowing for smaller, insulin-sensitive adipocytes. This cellular configuration minimizes hypoxia, reduces pro-inflammatory signaling, and preserves mitochondrial function. Conversely, impaired adipogenesis leads to adipocyte hypertrophy, tissue fibrosis, and chronic inflammation—all of which contribute to insulin resistance and systemic metabolic dysfunction. Thus, preserving or restoring functional adipogenesis is critical to preventing lipotoxicity and maintaining energy homeostasis [49]. Another promising strategy is the pharmacological activation of adipogenesis in APCs. One well-characterized approach is the activation of peroxisome proliferator-activated receptor gamma (PPARγ), a transcription factor essential for adipocyte differentiation. Agonists of PPARγ, such as thiazolidinediones, have been shown to enhance APC adipogenesis and improve insulin sensitivity in obese animal models. However, the long-term use of PPARγ agonists in humans is associated with adverse effects, including weight gain and fluid retention, which limits their clinical applicability [50]. Ongoing research is focused on developing more selective PPARγ modulators that can enhance adipogenesis without inducing unwanted side effects. In addition to PPARγ activation, bone morphogenetic protein 4 (BMP4) signaling has emerged as another key regulator of adipogenesis. BMP4 is a potent inducer of APC differentiation into adipocytes, and recent studies have suggested that small molecules or peptides that activate BMP4 signaling could promote adipogenesis and reverse fibrosis. Early-stage preclinical studies have shown promising results with BMP4 agonists, particularly in the context of visceral fat accumulation [51].

### 3.5. Cell-Based Therapies for Adipose Tissue Restoration

Cell-based therapies offer a more direct approach to enhancing APC function and restoring healthy adipose tissue. The transplantation of ex vivo expanded APCs or the genetic modification of endogenous APCs to enhance their adipogenic potential could offer long-term benefits for patients with obesity and insulin resistance. For example, recent studies explored the transplantation of adipose-derived stem cells (ADSCs) into obese animal models, showing improvements in adipose tissue function and metabolic health [52]. Gene editing technologies, such as CRISPR-Cas9, offer the possibility of enhancing the adipogenic potential of endogenous APCs [53]. By targeting genes involved in adipogenesis or fibrosis, such as C/EBPα or TGF-β receptors, it may be possible to restore the normal adipogenic program in APCs and reverse obesity-associated metabolic dysfunction [54].

### 3.6. Challenges and Future Directions

Despite the promising therapeutic potential of targeting APCs, several challenges remain. The heterogeneous nature of adipose progenitor cells and the complexity of their microenvironment pose significant obstacles to the development of effective therapies. Moreover, the long-term safety and efficacy of these interventions need to be carefully evaluated in clinical trials. Personalized approaches that consider the specific characteristics of individual patients, including their adipose tissue distribution and inflammatory profile, will likely be necessary for optimizing therapeutic outcomes [55].

In conclusion, targeting APCs holds significant potential for improving adipose tissue function and treating obesity-related metabolic diseases. By modulating APC differentiation, reducing fibrosis, and restoring tissue homeostasis, therapeutic strategies aimed at APCs may offer a promising approach for addressing the underlying causes of obesity and insulin resistance.

## 4. Adipose Tissue Inflammation and Insulin Resistance

Chronic low-grade inflammation is a hallmark of obesity and a pivotal mediator of insulin resistance. Among adipose depots, vWAT is particularly prone to inflammation, which is characterized by increased immune cell infiltration, elevated pro-inflammatory cytokines, and tissue remodeling. In contrast, sWAT exhibits a more protective inflammatory profile, particularly in the early stages of obesity, although this protection diminishes over time [56].

In vWAT, obesity induces the recruitment and activation of pro-inflammatory macrophages (M1 phenotype), T helper 1 (Th1) cells, and cytotoxic T cells, leading to the secretion of cytokines, such as tumor necrosis factor-alpha (TNF-α), interleukin-6 (IL-6), and monocyte chemoattractant protein-1 (MCP-1). These cytokines impair insulin signaling pathways in adipocytes by promoting the serine phosphorylation of insulin receptor substrates, reducing glucose uptake, and exacerbating systemic insulin resistance. In sWAT, although inflammation does occur, it tends to be less severe and may involve a higher proportion of anti-inflammatory immune cells, such as M2 macrophages and regulatory T cells (Tregs), which support tissue remodeling and insulin sensitivity [57].

Macrophages are central to the inflammatory landscape of adipose tissue. In lean states, tissue-resident M2 macrophages maintain homeostasis by clearing apoptotic cells and secreting anti-inflammatory cytokines, like IL-10. During obesity, however, there is a phenotypic switch to M1 macrophages, which form crown-like structures around dying adipocytes and release ROS and pro-inflammatory mediators that further impair adipocyte function. This inflammatory milieu is both a consequence and driver of mitochondrial dysfunction, as increased ROS and lipid accumulations disrupt mitochondrial homeostasis and fuel further cytokine production [58].

T cell subsets also play critical roles in shaping adipose inflammation. Th1 and CD8+ T cells promote inflammation and insulin resistance, while Th2 and Tregs exert protective effects. The balance between these subsets determines the degree of immune activation within each depot. Additionally, adipocytes themselves are not passive players; they secrete chemokines that recruit immune cells and upregulate MHC class II molecules, participating directly in antigen presentation and immune modulation [59].

Fibrosis, inflammation, and mitochondrial dysfunction are interlinked in the context of insulin resistance. The deposition of extracellular matrix components, particularly in vWAT, restricts the adipocyte expandability and exacerbates hypoxia, leading to further immune cell recruitment and inflammation. Mitochondrial dysfunction amplifies this cycle by increasing the ROS production and compromising lipid metabolism, leading to lipotoxicity in peripheral organs, such as the liver and muscle [60].

An emerging area of interest is the role of inflammation in thermogenic adipose depots—brown and beige adipose tissues (BAT and BeAT). While these tissues are typically anti-inflammatory and metabolically beneficial, recent evidence suggests that with chronic overnutrition or aging, BAT and BeAT can also undergo inflammatory remodeling. This inflammation may impair the thermogenic capacity and shift BAT toward a more WAT-like phenotype, potentially contributing to systemic insulin resistance. However, the precise conditions under which thermogenic fat contributes to or protects against inflammation remain incompletely understood [61,62].

In summary, adipose tissue inflammation is highly depot-specific and closely tied to insulin resistance. vWAT is a central node of inflammatory signaling and metabolic dysfunction, whereas sWAT and thermogenic depots may retain protective functions under certain conditions [63]. Understanding the cellular and molecular mediators of adipose inflammation offers critical insight into the pathogenesis of obesity-related metabolic disease and highlights potential therapeutic targets for restoring insulin sensitivity (Figure 2).

## 5. Mitochondrial Function and Metabolic Flexibility in Adipose Depots

Mitochondrial dysfunction is increasingly recognized as a central driver of adipose tissue impairment in obesity, contributing to systemic insulin resistance and metabolic disease. Adipose depots exhibit distinct mitochondrial characteristics, which influence their functional roles and responses to metabolic stress. In BAT, mitochondria are densely packed and specialized for non-shivering thermogenesis via uncoupling protein 1 (UCP1), which dissipates the proton gradient to generate heat instead of ATP. BeAT, found interspersed within sWAT, also exhibits inducible thermogenic capacity under certain stimuli, such as cold exposure or β-adrenergic activation [64,65].

In contrast, mitochondria in sWAT and vWAT primarily support lipid storage and metabolic flexibility—the ability to switch between glucose and fatty acid oxidation depending on the energy demand. This flexibility is vital for maintaining a systemic energy balance. However, in obesity, particularly in vWAT, the mitochondrial number, function, and plasticity are markedly reduced [66,67].

This fragmentation impairs oxidative phosphorylation efficiency, leading to decreased ATP production and enhanced generation of reactive oxygen species (ROS).

Excess ROS, in turn, activate redox-sensitive transcription factors, such as NF-κB and HIF-1α, promoting pro-inflammatory gene expression and exacerbating adipose tissue dysfunction [68]. ROS also damage mitochondrial DNA and proteins, further impairing the mitochondrial respiratory capacity and initiating a vicious cycle of oxidative stress and inflammation, as seen in Figure 3. Moreover, defective mitophagy, characterized by the impaired clearance of dysfunctional mitochondria, has been documented in obese adipose tissue, particularly in vWAT [69,70].

A reduced expression of key autophagy regulators, such as PINK1 and Parkin, leads to the accumulation of dysfunctional mitochondria, thereby reinforcing oxidative stress and cellular dysfunction. In contrast, brown adipose tissue (BAT) initially maintains relatively preserved mitochondrial integrity and thermogenic function, primarily through sustained uncoupling protein 1 (UCP1) activity. However, under prolonged metabolic stress, the mitochondrial structure and function within BAT deteriorate, resulting in impaired thermogenesis and contributing to a systemic energy imbalance [71,72]. Thus, the loss of mitochondrial quality control mechanisms, particularly in vWAT, represents a critical contributor to adipose tissue dysfunction, insulin resistance, and the progression of metabolic disease in obesity.

Obesity is associated with a fragmented mitochondrial morphology, decreased mitochondrial membrane potential, impaired oxidative phosphorylation, and reduced expression of key regulators of mitochondrial biogenesis, such as peroxisome proliferator-activated receptor gamma coactivator-1α (PGC-1α) [73]. These alterations contribute to metabolic inflexibility, lipid spillover, and ectopic fat accumulation [74].

Depot-specific mitochondrial dynamics further influence local insulin sensitivity. In vWAT, mitochondrial dysfunction is tightly linked to the development of insulin resistance through mechanisms involving increased reactive oxygen species (ROS) production and disrupted fatty acid oxidation. The accumulation of ROS not only damages mitochondrial DNA and proteins but also activates inflammatory pathways that exacerbate tissue dysfunction. Conversely, sWAT shows a comparatively preserved mitochondrial function in the early stages of obesity, although prolonged metabolic stress eventually impairs its metabolic adaptability as well [75,76].

Autophagy and mitophagy—key processes in mitochondrial quality control—are also disrupted in obesity, particularly in vWAT. The accumulation of dysfunctional mitochondria due to impaired mitophagy promotes cellular stress, inflammation, and a decline in adipocyte function [77]. Notably, in thermogenic depots, mitochondrial integrity is essential for adaptive responses to environmental and nutritional cues. A decline in mitochondrial function in BAT and BeAT not only reduces the thermogenic capacity but may also contribute to systemic metabolic impairment, especially in aging or obesity [78]. Thus, mitochondrial health across adipose depots plays a pivotal role in determining adipose tissue plasticity, insulin sensitivity, and overall metabolic homeostasis. Targeting mitochondrial biogenesis, dynamics, and oxidative stress pathways represents a promising therapeutic avenue to restore metabolic flexibility and counteract the deleterious effects of obesity.

## 6. Adipokines and Systemic Metabolic Regulation

Adipose tissue acts as a dynamic endocrine organ, secreting a variety of bioactive molecules known as adipokines, which regulate the energy balance, the glucose and lipid metabolisms, inflammation, and the insulin sensitivity [49,79]. Among the best-characterized adipokines, leptin and adiponectin have crucial and complementary roles. Leptin primarily regulates appetite and energy expenditure via central nervous system pathways [80], whereas adiponectin enhances the peripheral insulin sensitivity, exerts anti-inflammatory effects, and promotes cardiovascular protection [81]. In lean individuals, subcutaneous white adipose tissue (sWAT) is the predominant source of adiponectin, supporting insulin sensitivity and metabolic flexibility [82].

Conversely, visceral white adipose tissue (vWAT) predominantly secretes pro-inflammatory adipokines, including resistin and visfatin, both of which contribute to systemic insulin resistance and inflammation [83,84]. In obesity, vWAT undergoes pathological expansion and becomes a significant source of inflammatory cytokines, including tumor necrosis factor-alpha (TNF-α) and interleukin-6 (IL-6), exacerbating systemic insulin resistance [85].

At the same time, adiponectin levels decline, particularly from sWAT, contributing to impaired lipid metabolism, oxidative stress, and ectopic fat deposition [86]. Beyond white adipose depots, brown adipose tissue (BAT) and beige adipose tissue (BeAT) also secrete specialized endocrine factors called batokines.

Notable batokines include fibroblast growth factor 21 (FGF21), which enhances glucose uptake and lipid oxidation [87,88], neuregulin 4 (NRG4), which protects against hepatic steatosis [89], and bone morphogenetic protein 8b (BMP8b), which enhances the thermogenic capacity [90]. Thus, BAT and BeAT not only contribute to thermogenesis but also influence systemic glucose and lipid metabolism through endocrine communication. Adipokine-mediated inter-organ communication is central to systemic metabolic regulation. Leptin acts on the hypothalamus to modulate appetite and energy expenditure [91], while adiponectin enhances the hepatic insulin sensitivity by promoting fatty acid oxidation and suppressing gluconeogenesis [71]. In skeletal muscle, adiponectin promotes glucose uptake and mitochondrial biogenesis, improving insulin responsiveness [92]. Conversely, elevated levels of pro-inflammatory adipokines impair pancreatic β-cell survival and insulin secretion, exacerbating the development of type 2 diabetes [93]. In obesity, this finely tuned network becomes disrupted. Chronic exposure to high levels of inflammatory adipokines leads to low-grade systemic inflammation, metabolic inflexibility, and a vicious cycle of insulin resistance [85]. Given the depot-specific patterns of adipokine secretion, targeted therapeutic strategies are being explored [94].

Potential approaches include restoring adiponectin activity, for example, through adiponectin receptor agonists [95]; suppressing inflammatory adipokines by modulating adipose immune cell profiles [96]; and enhancing batokine signaling, particularly with FGF21 analogs or neuregulin 4-based therapies [97]. Several of these interventions are currently being evaluated in clinical trials for the treatment of obesity-related metabolic disorders. Thus, modulating adipokine networks with a depot-specific strategy holds significant promise for restoring metabolic homeostasis and improving systemic insulin sensitivity. Emerging therapeutic strategies targeting adipokine pathways are under intense investigation to counteract obesity-related metabolic dysfunction. Adiponectin receptor agonists, such as AdipoRon, have shown promising effects in preclinical models by mimicking adiponectin’s insulin-sensitizing and anti-inflammatory actions [95].

Similarly, fibroblast growth factor 21 (FGF21) analogs, such as pegbelfermin, have progressed into clinical development, showing beneficial effects on lipid metabolism, insulin sensitivity, and hepatic steatosis in obese individuals with type 2 diabetes [98].

Moreover, targeting neuregulin 4 (NRG4), a batokine with protective effects against hepatic lipid accumulation, represents a novel approach to restoring metabolic balance. Although NRG4-targeted therapies are still in preclinical stages, recent data highlight its potential as a regulator of systemic glucose and lipid metabolism [99].

Figure 4 illustrates the systemic effects of adipokines secreted from distinct adipose depots.

Thus, the modulation of adipokine signaling represents a promising therapeutic avenue, with several agents currently in preclinical or clinical development aimed at restoring systemic metabolic homeostasis.

## 7. Therapeutic Implications and Future Directions

As the heterogeneity of adipose tissue depots becomes increasingly clear, it is evident that tailored, depot-specific therapeutic strategies are necessary to address the multifaceted nature of obesity-related metabolic dysfunction [100]. Interventions aimed at selectively activating BAT or BeAT, reducing vWAT inflammation, and mitigating fibrosis in sWAT may hold the key to more precise and effective treatments [101].

Stimulating BAT and BeAT thermogenesis offers a promising route for increasing energy expenditure and improving glucose homeostasis [102]. Pharmacological agents, such as β3-adrenergic receptor agonists, capsaicin analogs, and thyroid hormone mimetics, have shown the ability to enhance BAT activity and induce the browning of sWAT in animal models and early human trials [103]. However, challenges remain in translating these effects into sustained metabolic benefits in obese, insulin-resistant individuals, where BAT functionality is often compromised [104]. Moreover, the long-term safety of systemic thermogenic activation, particularly concerning cardiovascular side effects, requires careful evaluation [105].

In contrast, vWAT represents a major source of metabolic inflammation. Therapeutic approaches targeting vWAT have focused on immune modulation, with the aim of restoring a more anti-inflammatory immune cell composition. Agents that promote M2 macrophage polarization or expand regulatory T cell (Treg) populations—such as IL-10 mimetics, PPARγ modulators, and TGF-β inhibitors—have shown promise in reducing local cytokine levels and improving insulin sensitivity [106,107]. Additionally, nutritional strategies, including omega-3 fatty acids and polyphenols, may dampen vWAT inflammation through their immunomodulatory properties [108].

Fibrotic remodeling in sWAT is another barrier to healthy adipose expansion and function. Inhibiting pathways, such as TGF-β/SMAD signaling or lysyl oxidase-mediated collagen crosslinking, may reduce the ECM stiffness and promote adipogenic differentiation of APCs. Nutritional interventions (e.g., high-fiber diets, resveratrol), exercise, and agents enhancing ECM turnover (such as matrix metalloproteinases activators) are under investigation for their antifibrotic potential [100].

Among the most exciting avenues of research is the modulation of adipose progenitor cells (APCs) to favor adipogenesis over fibrogenesis. Strategies include the pharmacological stimulation of PPARγ or BMP4 pathways; gene editing (e.g., targeting C/EBPα or TGF-β receptors); and cell-based therapies, such as the transplantation of ex vivo expanded or genetically modified APCs. However, a deeper understanding of depot-specific APC behavior and microenvironmental cues is essential before clinical application [84,109,110].

Sexual dimorphism significantly influences the adipose tissue distribution and function and the pathophysiology of obesity. Estrogen signaling enhances insulin sensitivity, promotes mitochondrial biogenesis, and upregulates anti-inflammatory pathways in subcutaneous adipose tissue, partly via the ERα-mediated activation of AMPK and PGC-1α. In contrast, androgen excess in females (e.g., PCOS) or testosterone deficiency in males is associated with increased visceral adiposity and metabolic dysregulation. Sex differences also extend to adipocyte progenitor proliferation, immune cell infiltration (e.g., higher M1/M2 macrophage ratio in males), and the differential browning capacity of white adipose depots. These mechanisms collectively shape sex-specific susceptibility to insulin resistance and therapeutic response profiles [111,112].

Despite the progress, important questions remain: Can BAT activation be leveraged effectively in patients with severe insulin resistance and dysfunctional thermogenic tissue? What are the long-term effects of immunomodulatory therapies on adipose tissue plasticity and metabolic resilience? Can we selectively reprogram APCs without compromising their regenerative or immune-regulatory roles? [100].

Future research must integrate genomics, epigenetics, and adipose tissue engineering to develop personalized and sustainable interventions. Single-cell transcriptomic and spatial omics approaches will offer detailed maps of adipose cell populations, while organoid models and bioprinted adipose tissue constructs will enable mechanistic exploration and drug testing in physiologically relevant contexts [101].

Understanding the depot-specific roles of adipose tissue in metabolic regulation reveals the intricate interplay between inflammation, mitochondrial dysfunction, fibrosis, and endocrine signaling in the progression from metabolic health to insulin resistance. Far from being a passive energy reservoir, adipose tissue is a dynamic and heterogeneous organ that can either support or sabotage systemic homeostasis. Targeting distinct adipose depots offers promising avenues for precision therapies in obesity-related diseases. Key strategies include restoring mitochondrial integrity, modulating immune responses, and reversing fibrotic remodeling. Unraveling this complexity is not only crucial for developing effective interventions but also for redefining our approach to treating metabolic disorders from a tissue-specific perspective.

In conclusion, advancing therapeutic strategies that account for adipose depot heterogeneity—while embracing cutting-edge technologies—will be critical for combating obesity and its complications. A multidisciplinary approach that bridges molecular insight with clinical innovation may ultimately reshape how we treat metabolic disease.

### Clinical Applications and Translational Perspectives

The dysfunction of adipose tissue in obesity has catalyzed the development of targeted therapeutic strategies aimed at restoring tissue homeostasis and systemic metabolic balance. Given the critical roles of adipokines, mitochondrial health, fibrosis, and adipose progenitor cell (APC) function in metabolic regulation, a number of preclinical and clinical interventions have been proposed. Several emerging therapies targeting adipose tissue dysfunction are currently in preclinical or clinical development. These include PPARγ agonists (e.g., pioglitazone [113]), adipokine pathway modulators (such as AdipoRon [95], pegbelfermin [98], and NRG4 analogs [99]), fibrosis-targeting agents (e.g., fresolimumab [114]), and mitochondrial function enhancers (e.g., elamipretide [115]). Collectively, these interventions aim to improve insulin sensitivity, reduce fibrosis and inflammation, and restore mitochondrial integrity. PPARγ agonists, such as pioglitazone, have been shown to improve insulin sensitivity and promote adipocyte differentiation through the high-affinity activation of PPARγ [113]. However, their clinical use is limited by adverse effects, including weight gain and fluid retention, as reported in subsequent clinical studies. Although the translation of these therapeutic approaches into clinical practice remains in its early stages, the progress observed in preclinical and early-phase clinical trials offers promising avenues for mitigating the metabolic complications associated with obesity. Future investigations that focus on optimizing efficacy, minimizing side effects, and ensuring long-term safety are essential for the successful implementation of these strategies in clinical settings.

Table 2 summarizes key clinical and preclinical therapeutic interventions targeting adipose tissue dysfunction, fibrosis, mitochondrial impairment, and adipokine signaling.

Building on these therapeutic advances, further understanding of depot-specific roles of adipose tissue will reveal new therapeutic opportunities. In conclusion, understanding the depot-specific roles of adipose tissue in metabolic health reveals new therapeutic opportunities. Precision strategies that restore mitochondrial function, modulate immune responses, and reverse fibrotic remodeling may ultimately transform the management of obesity and its associated complications. A multidisciplinary and translational approach will be essential for designing effective, sustainable interventions.

## Figures and Tables

**Figure 1 cimb-47-00343-f001:**
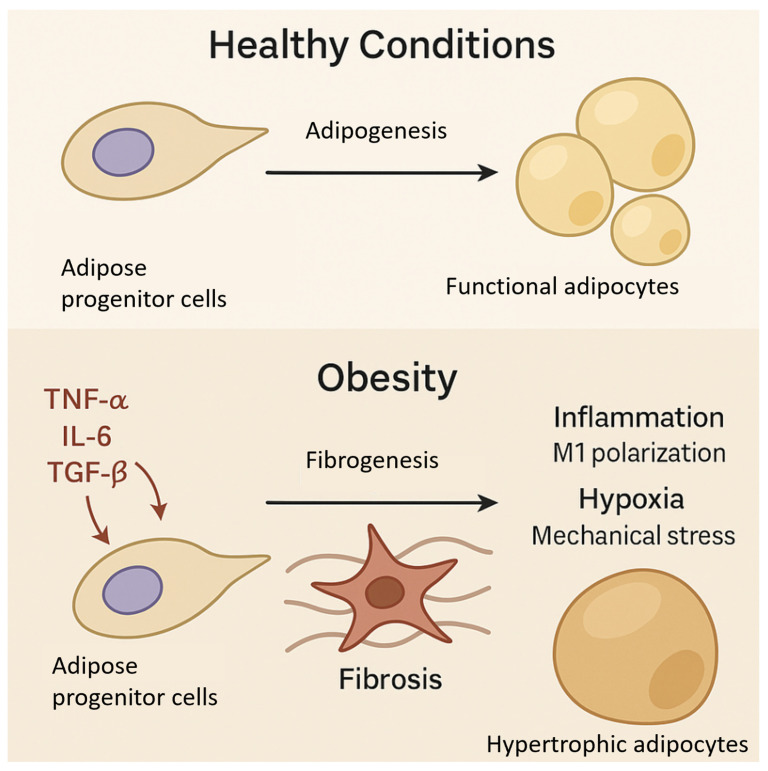
Schematic representation of adipose progenitor cell (APC) differentiation under healthy and obese conditions. Under healthy metabolic conditions, APCs predominantly differentiate into functional adipocytes through adipogenesis, supporting normal adipose tissue expansion and lipid storage. In obesity, chronic exposure to pro-inflammatory cytokines (TNF-α, IL-6) and fibrotic signals (TGF-β), along with hypoxia and mechanical stress, shifts APC differentiation toward fibrogenesis. This maladaptive response leads to fibrosis, reduced tissue plasticity, hypertrophic adipocyte formation, local inflammation (via M1 macrophage polarization), and impaired metabolic function.

**Figure 2 cimb-47-00343-f002:**
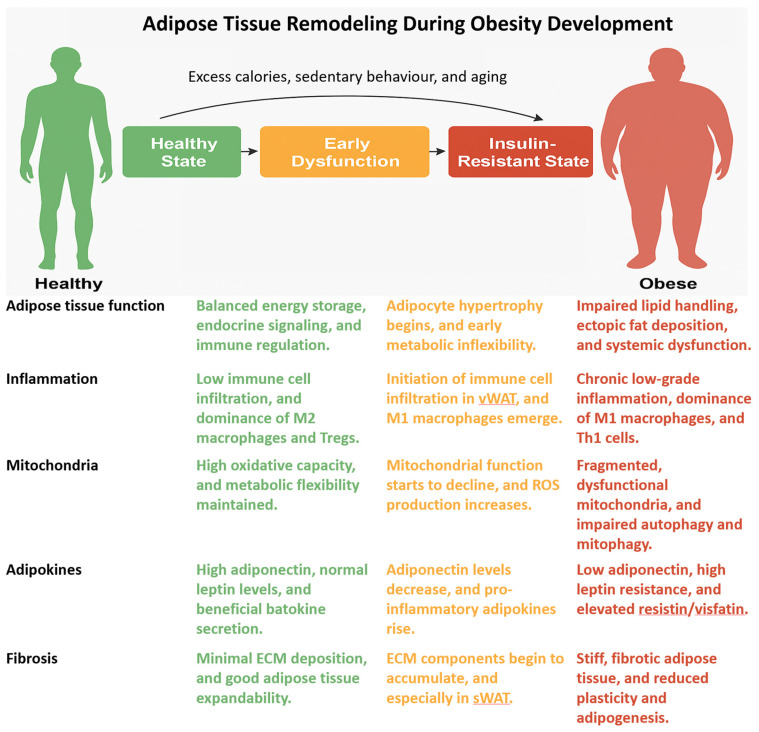
Progressive alterations in adipose tissue depots during the transition from metabolic health to insulin resistance. Each depot undergoes distinct immune, mitochondrial, and fibrotic changes that contribute to systemic metabolic dysfunction.

**Figure 3 cimb-47-00343-f003:**
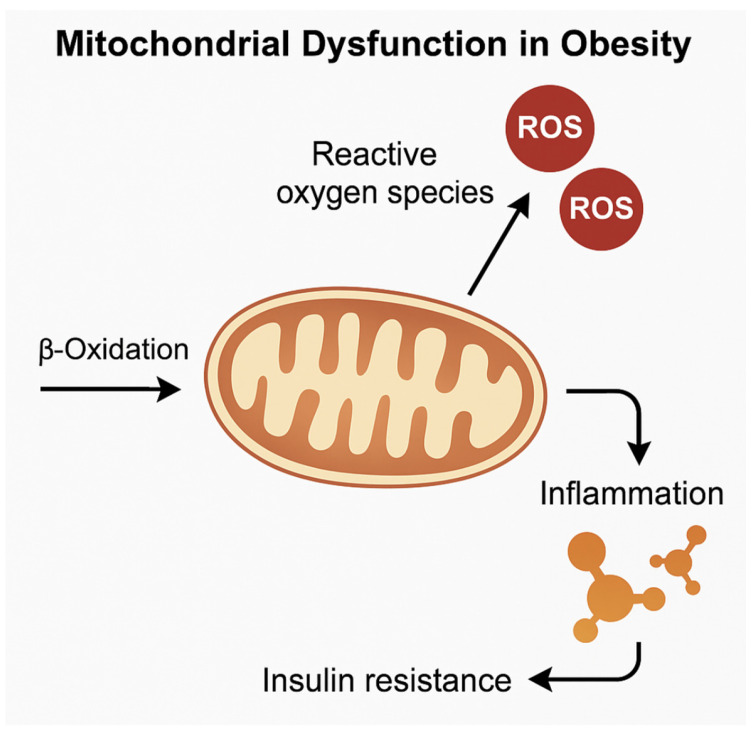
Obesity impairs mitochondrial function in adipose depots, promoting ROS production, inflammation, loss of metabolic flexibility in vWAT, and reduced thermogenesis in BAT and BeAT.

**Figure 4 cimb-47-00343-f004:**
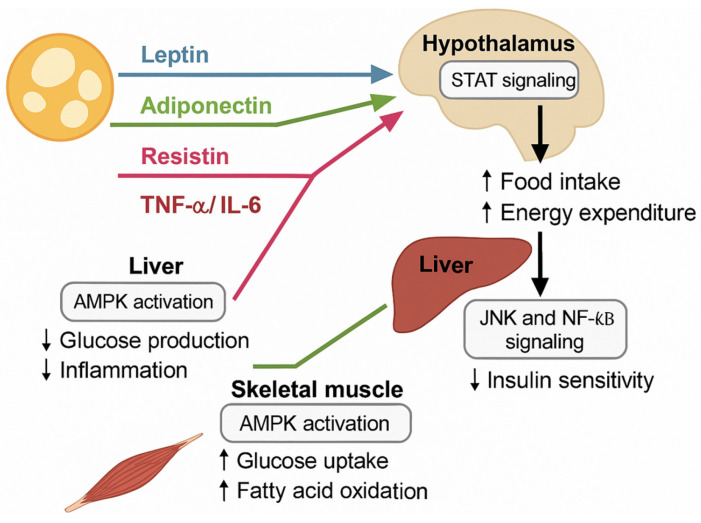
Adipokines and systemic metabolic regulation. Adipose tissue-derived adipokines exert depot-specific effects on major metabolic organs. Leptin and adiponectin enhance hypothalamic STAT signaling to regulate food intake and energy expenditure and promote AMPK activation in the liver and skeletal muscle, leading to reduced hepatic glucose production, decreased inflammation, increased glucose uptake, and fatty acid oxidation. Conversely, pro-inflammatory adipokines such as resistin, TNF-α, and IL-6, impair the metabolic function by activating JNK and NF-κB pathways in the liver, decreasing the insulin sensitivity and exacerbating systemic inflammation.

**Table 1 cimb-47-00343-t001:** Overview of the key features of different adipose tissue depots.

Brown Adipose Tissue (BAT)	Beige Adipose Tissue (BeAT)	Subcutaneous White Adipose Tissue (sWAT)	Visceral White Adipose Tissue (vWAT)
High mitochondria density, UCP1+	Induced in sWAT (transdifferentiation)	Metabolically protective (lean)	Pro-inflammatory (M1 macrophages)
Thermogenesis	Variable thermogenic capacity	Fibrosis under metabolic stress	High lipolytic activity
Improves glucose metabolism	Plasticity	APC-mediated adipogenesis	APC heterogeneity affects remodeling
Sensitive to aging/obesity	Sensitive to inflammatory signals	Single-nucleus RNA-seq data advances	Associated with insulin resistance
APC role emerging	May become pro-inflammatory		

Brown adipose tissue (BAT) is highly thermogenic due to abundant mitochondria and UCP1 expression, playing a protective metabolic role. Beige adipose tissue (BeAT) arises from subcutaneous white adipose tissue (sWAT) through transdifferentiation and exhibits metabolic flexibility, although it may become dysfunctional under obesogenic stress. sWAT, while generally protective, undergoes fibrosis and immune infiltration in metabolic disease states, with adipose progenitor cells (APCs) playing a crucial role in maintaining homeostasis. Visceral white adipose tissue (vWAT) is strongly associated with inflammation, fibrosis, and systemic insulin resistance, with both immune and progenitor cells contributing to its pathological remodeling.

**Table 2 cimb-47-00343-t002:** Therapeutic strategies targeting adipose tissue dysfunction in obesity.

Molecule/Therapy	Target Pathway	Clinical Status	Main Effect
Pioglitazone	PPARγ activation	Approved	Improves insulin sensitivity
Pegbelfermin	FGF21 analog	Phase II trials	Enhances lipid and glucose metabolism
AdipoRon	Adiponectin receptor agonist	Preclinical	Mimics adiponectin’s insulin-sensitizing effects
Fresolimumab	TGF-β neutralization	Phase II trials	Reduces fibrosis and inflammation
Elamipretide	Mitochondrial membrane stabilization	Phase II trials	Restores mitochondrial function

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
