# Peer review of "Key Roles of Brown, Subcutaneous, and Visceral Adipose Tissues in Obesity and Insulin Resistance"

_cimb, 2025, doi:10.3390/cimb47050343_

Round 1
Reviewer 1 Report
Comments and Suggestions for Authors
Dobre et al present an overall well-written review on a vast, topic of increasing current interest*.
The content stays mainly general, lacking often mechanistic details. The selected references refer mainly to reviews. More regretfully, in some chapters such as chapter 2, many references are badly chosen, with little relevance to the treated subject. Many landmark papers are not cited. The distinction between clinical studies and observations made in animal models could be stated more clearly.
Insulin resistance should be more clearly defined.
The important information in figure 1 is illegible. The idea is interesting, but as such the figure is not acceptable.
*More than 830 reviews on "adipose tissue", obesity and "insulin resistance" have been published within the last 5 years!
Specific remarks :
Line 31 : Currently, the term MAFLD is more commonly used than NAFLD.
Chap 2.1 : At least one of the 3 papers in N Engl J Med from 2009 describing BAT in humans ought to be cited along with the Becher (Nat Med) paper from 2021.
Chap 2.3 : The main part of this chapter focus on progenitor cells and single cell RNA seq studies not specifically on sWAT. The authors should either change the title or the content of the paragraph. If the latter, the specificity of sWAT must be emphasized.
Recent years many interesting studies on single cells have been published. Among them, the study of Emont (Nature 2022) deserves being cited as well as one from Massier (Nat Comm 2023) since it contains a meta-analysis of previous studies. The study of Bäckdahl (Cell Metab) from 2021 is also informative. Along with Emont et al, Gupta (Genome Res 2022) points out the usefulness of performing single nuclei seq (compared to scRNA-seq) when studying the transcriptome of adipocytes.
Lines 108-111 : Hildreth et al might not be the most appropriate ref.
Lines 119-121: It seems as the authors mixed up SWAT (structural Wnt-regulated adipose tissue-resident) cells (in Palini, ref23) with their definition of sWAT. SWAT cells do not differentiate into adipocytes. The paper does not study any inflammatory mechanisms.
Ref 24 is also badly chosen.
Chap 3.4 : The beneficial effects of adipogenesis could be more clearly developed.
Other comments
Affiliation #2 has not been attributed
Aspects concerning sexual dimorphism of obesity are missing. This subject should maybe be considered, being an emerging area of interest.
Reviewer 2 Report
Comments and Suggestions for Authors
This review highlights the various factors contributing to obesity and is generally well-organized with a logical flow. Overall, it presents a good foundation. However, revisions are necessary before it can be considered for acceptance. Below are some suggestions and comments:
- Introduction Structure: The introduction could be better structured by dividing it into three parts: (1) a brief overview of obesity as a disease, (2) a summary of the key contributing factors, and (3) an outline of the main themes covered in the review. Important concepts discussed later in the manuscript—such as APC and mitochondria—should be briefly introduced in this section to provide context and guide the reader.
- Figures and Visual Aids: The manuscript currently includes only one figure, which is insufficient for a comprehensive review. Additional figures are needed, especially for Parts 2, 3, and 5, where complex mechanisms are discussed. Each figure should be accompanied by a detailed legend to enhance clarity and understanding.
- Section 6 - Adipokines and Systemic Metabolic Regulation: This section discusses a highly complex aspect of obesity involving multiple signaling pathways. More in-depth analysis is encouraged. Including a well-designed figure summarizing these pathways would significantly aid in conveying the information.
- Abbreviations: Abbreviations should be defined only at their first appearance in the text. Subsequent uses should use the abbreviation alone, without repeating the full term.
- Scope and Completeness: While the review seems to aim at providing a systematic overview of obesity and insulin resistance, several sections lack completeness and depth. Expanding on underdeveloped areas would strengthen the manuscript.
- Clinical Relevance: The review lacks discussion of clinical applications, including drugs or small molecules that target the pathways discussed. It is recommended to include a systematic table summarizing therapeutic interventions and to add a dedicated paragraph discussing relevant clinical trials and preclinical studies.
It is fine but can be improved.
Round 2
Reviewer 1 Report
Comments and Suggestions for Authors
The manuscript has clearly improved in the revised version. Not all modifications had been colored in red but I figure this was deliberate to improve lisibility.
I think it was a good idea to add the new figures to support certain text sections.
I believe the authors meant to cite PMID: 35042723 in ref #26 instead of the cited ref.
I have no further comment.
Author Response
We sincerely thank the reviewer for the positive feedback and for acknowledging the improvements made in the revised version of the manuscript.
We confirm that not all changes were marked in red, as we aimed to maintain clarity and readability throughout the revised text.
We appreciate the suggestion regarding reference #26. Indeed, our intention was to cite PMID: 35042723, and we have now corrected the reference accordingly.
Thank you once again for your valuable input and support in improving the quality of our manuscript.
Reviewer 2 Report
Comments and Suggestions for Authors
I agree to accept it as the author did revisions according to my comments.
Author Response
We thank the reviewer for the time and effort dedicated to evaluating our manuscript. We are grateful for the constructive comments and are pleased that the revised version meets your expectations.